# Insights from a Computational-Based Approach for Analyzing Autophagy Genes across Human Cancers

**DOI:** 10.3390/genes14081550

**Published:** 2023-07-28

**Authors:** Alexis Germán Murillo Carrasco, Guilherme Giovanini, Alexandre Ferreira Ramos, Roger Chammas, Silvina Odete Bustos

**Affiliations:** 1Center for Translational Research in Oncology (LIM24), Instituto do Cancer do Estado de Sao Paulo (ICESP), Hospital das Clinicas da Faculdade de Medicina da Universidade de Sao Paulo (HCFMUSP), São Paulo 01246-000, Brazil; agmurilloc@usp.br (A.G.M.C.); silvina.bustos@hc.fm.usp.br (S.O.B.); 2Comprehensive Center for Precision Oncology, Universidade de São Paulo, São Paulo 01246-000, Brazil; 3Escola de Artes, Ciências e Humanidades, Universidade de São Paulo, Av. Arlindo Béttio, 1000, São Paulo 03828-000, Brazil; ggiovanini@usp.br (G.G.); alex.ramos@usp.br (A.F.R.)

**Keywords:** systems biology, autophagy, cancer, cell phenotype, cancer dataset

## Abstract

In the last decade, there has been a boost in autophagy reports due to its role in cancer progression and its association with tumor resistance to treatment. Despite this, many questions remain to be elucidated and explored among the different tumors. Here, we used omics-based cancer datasets to identify autophagy genes as prognostic markers in cancer. We then combined these findings with independent studies to further characterize the clinical significance of these genes in cancer. Our observations highlight the importance of innovative approaches to analyze tumor heterogeneity, potentially affecting the expression of autophagy-related genes with either pro-tumoral or anti-tumoral functions. In silico analysis allowed for identifying three genes (*TBC1D12*, *KERA*, and *TUBA3D*) not previously described as associated with autophagy pathways in cancer. While autophagy-related genes were rarely mutated across human cancers, the expression profiles of these genes allowed the clustering of different cancers into three independent groups. We have also analyzed datasets highlighting the effects of drugs or regulatory RNAs on autophagy. Altogether, these data provide a comprehensive list of targets to further the understanding of autophagy mechanisms in cancer and investigate possible therapeutic targets.

## 1. Introduction

Autophagy, a conserved process, plays a vital role in regulating cell metabolism and homeostasis in both physiological and pathological circumstances [1]. By recycling nutrients and amino acids, autophagy contributes to metabolic adaptation in cancer cells, which can either facilitate cancer progression or induce cell death, depending on the stage of the disease. Autophagy-related genes (ATGs) are key regulators involved in the complex mechanism of autophagy [1]. Different types of autophagy, such as macroautophagy, chaperone-mediated autophagy (CMA), and microautophagy have been extensively described.

Macroautophagy involves the formation of autophagosomes, double-layered membranes that sequester cellular cargo and subsequently fuse with lysosomes for degradation [2]. Nutrient deprivation and stress regulate macroautophagy, which can be inhibited by the mechanistic target of rapamycin (mTOR), a master growth regulator, and activated by AMP-activated protein kinase (AMPK) and the hypoxia-inducible transcription factor (HIF), both of which are involved in stress response pathways. These regulatory factors collectively contribute to autophagic degradation and the maintenance of cellular homeostasis [3,4].

Chaperone-mediated autophagy is a selective mechanism that relies on specialized molecular chaperones to recognize and deliver specific proteins to the lysosomes for degradation. The chaperone heat shock cognate 70 (Hsc70) plays a crucial role in this process, as it recognizes a specific amino acid sequence motif, called the KFERQ consensus motif, present in target proteins. Upon binding to a protein containing the KFERQ motif, Hsc70 forms a complex with co-chaperones, facilitating the unfolding of the protein and its translocation across the lysosomal membrane. Upon reaching the lysosomal membrane, the complex interacts with a lysosomal receptor called lysosome-associated membrane protein type 2A (LAMP-2A), which aids in the translocation of the target protein into the lysosomal lumen for degradation by lysosomal enzymes [5].

Mitophagy is a cellular process that selectively removes damaged mitochondria through autophagy, thereby preserving mitochondrial fidelity by degrading and replacing damaged mitochondria. When mitochondria are damaged, PTEN-induced putative kinase 1 (PINK1) accumulates on the outer mitochondrial membrane and, after phosphorylation, facilitates binding to substrates ubiquitin and Parkin (PARK2). The ubiquitinated mitochondria are recognized by Sequestosome 1 (p62/SQSTM1), a cargo receptor protein, that interacts with microtubule-associated protein 1 light chain 3 (LC3), a component of the autophagosomal membrane [6].

Microautophagy is a non-selective lysosomal degradative process wherein the lysosomal membrane invaginates or protrudes to directly engulf small portions of the cytoplasm or specific organelles [7].

Several cargo-specific autophagy processes have also been reported, including peroxisome removal (pexophagy), endoplasmic reticulum degradation (erphagy/reticulophagy), ribosome degradation (ribophagy), lipid droplet degradation (lipophagy), elimination of invading microbes (xenophagy), clearance of protein aggregates (aggrephagy), and degradation of nuclear material (nucleophagy) [8].

Given the significance of autophagy in cancer, we conducted an analysis of autophagy-related genes in various tumor tissues associated with different types of cancers. The analyzed tumor tissues included bladder urothelial carcinoma (BLCA), breast invasive carcinoma (BRCA), cholangiocarcinoma (CHOL), colon adenocarcinoma (COAD), esophageal carcinoma (ESCA), glioblastoma multiforme (GBM), head and neck squamous cell carcinoma (HNSC), kidney chromophobe (KICH), kidney renal clear cell carcinoma (KIRC), kidney renal papillary cell carcinoma (KIRP), brain lower grade glioma (LGG), liver hepatocellular carcinoma (LIHC), lung adenocarcinoma (LUAD), lung squamous cell carcinoma (LUSC), ovarian serous cystadenocarcinoma (OV), prostate adenocarcinoma (PRAD), rectum adenocarcinoma (READ), skin cutaneous melanoma (SKCM), stomach adenocarcinoma (STAD), testicular germ cell tumors (TGCT), thyroid carcinoma (THCA), uterine corpus endometrial carcinoma (UCEC), colorectal adenocarcinoma (COAD + READ or COADREAD), pan-kidney cohort (KICH + KIRC + KIRP or KIPAN), and stomach and esophageal carcinoma (STAD + ESCA or STES).

In this comprehensive review, we compiled and analyzed autophagy-related gene expression data using multiple bioinformatics approaches complemented by reviewing independent literature sources to validate and corroborate our findings. By combining these methodologies, we aimed to enhance our understanding of autophagy pathways and their involvement in the development and progression of cancer.

## 2. Methods

### 2.1. Study Databases

To explore the role of autophagy genes in cancer, we adopted a systematic approach. Initially, we utilized the Molecular Signature Database v2023.1.Hs (MSigDB released on March 2023, [9]) in the modules Curated Gene Sets (C2) and Oncology Gene Sets (C5) to obtain a list of 707 genes associated with all available types of autophagy (Appendix A). Although other datasets could be available for autophagy-related genes [10,11], our strategy was to choose a more balanced and updated source between macroautophagy- and microautophagy-related processes. As autophagy research is still developing, especially in tumors, we propose a robust approach using robust statistical criteria and evaluating different methods to compare gene expression levels, which may be further strengthened/verified as more information is published. Subsequently, we acquired gene expression data from The Cancer Genome Atlas (TCGA) project for evaluating the expression levels of these autophagy-related genes. To ensure comparability, we augmented the tissue datasets with information from the Genotype-Tissue Expression (GTEx) and Therapeutically Applicable Research to Generate Effective Treatments (TARGET) projects, thereby attaining a balanced number of samples across organs.

### 2.2. Analytical Approaches for Evaluating Autophagy-Related Genes in Tumors 

To evaluate the expression profile of autophagy-related genes in an unbiased method, we performed a differential expression analysis followed by a clusterization by Uniform Manifold Approximation and Projection (UMAP), as shown in Appendix A. We used the Xena repository at the University of California Santa Cruz (UCSC, [12]) for retrieving expression data of TCGA + GTEx + TARGET cohort and the FirebrowseR package [13] for the TCGA dataset.

For differential expression analysis, we evaluated three approaches: Approach A compares the expression levels between tumor and normal-adjacent samples (only available in the TCGA cohort) using the Mann–Whitney test; Approach B evaluates differentially expressed genes in a large cohort of samples (tumor vs. normal) from the TCGA + GTEx + TARGET cohort using the Mann–Whitney test; and Approach C compares matched samples between normal-adjacent and tumor samples using paired Wilcoxon’s test.

These approaches were designed considering that: (i) TCGA is a large patient cohort of tumor samples previously used for different gene-based analyses, even of autophagy-related genes [14]; (ii) it is possible to increase the statistical power of this analysis by complementing TCGA data with GTEx and TARGET cohorts for normal and tumor samples, respectively; and (iii) although it limits the overall sample size, a paired analysis could increase the statistical power in a different way [15].

### 2.3. Tumor Clusterization Based on Autophagy-Related Genes

After evaluating common differential genes between these approaches, we continue with the large dataset (TCGA + GTEX + TARGET, Approach B) using fold change between tumor and normal samples to cluster all tumors depending on autophagy-related factors. 

For this analysis, we included all tissues where the fold changes between tumor and normal samples were estimated using at least 100 participants per group. Then, we considered each tissue (*n* = 16) as a representative value that includes the following tissues: BRCA, COAD, ESCA, GBM, KIRC, KIRP, LGG, LIHC, LUAD, LUSC, OV, PAAD, PRAD, SKCM, STAD, and TGCT. Due to the number of tissues, we ran a principal component analysis (PCA) followed by a UMAP using eight dimensions. Then, we included all autophagy-related genes minimally expressed in the 50% of participant tissues per cluster with a fold change threshold above 1.5 as cluster-associated markers.

Finally, to evaluate the specific contribution between clusters 0 + 1 vs. 2, we included all genes differentiating clusters 0 and 1 in a random forest Gini importance analysis by simulating one hundred thousand trees to estimate mean decrease accuracy and mean decrease Gini values.

## 3. Results and Discussion 

### Autophagy-Related Genes Differentially Expressed in Solid Tumors

Initially, we plotted heatmaps (Figure 1) considering three different approaches for the analysis of differentially expressed genes: (A) the fold in the median expression levels of normal and tumor tissues among TCGA participants; (B) the fold in the median expression levels of normal and tumor tissues among participants in TCGA + GTEX + TARGET; and (C) the median of folds matched by TCGA participants.

These three approaches allowed us to identify commonly overexpressed genes in the cohort of solid tumors analyzed. Notable genes included *TOP2A*, *CENPK*, *TUBB3*, *TRIB3*, *TUBA1C*, and *MET* (except in PRAD and BRCA), as well as *TREM2* (except in LUSC and LUAD), which exhibited underexpression. Specifically, approaches A and C revealed overexpression of *RIPK2* and *LAMP3* (except in LUSC and LUAD), while Approach B identified overexpression of *GAPDH* and *GRAMD1A*. Additionally, *TP53* and *FRMD5* were found to be overexpressed in Approach B but exhibited underexpression in KIRC, LIHC, PRAD, and TGCT. *CDK5R1* showed exclusive overexpression in Approach C. 

A discussion of some of the key findings follows.

TOP2A (topoisomerase II α) is an enzyme involved in DNA topology rearrangements, and its aberrant expression is linked to various cancer types. It serves as a target for anticancer drugs like Doxorubicin and etoposide, which have been associated with autophagy promotion [16,17]. Amplification and deletion of *TOP2A* are associated with both sensitivity and resistance to topoII-inhibitor-based chemotherapy [18].CENPK (centromere protein K) is a component of the centromeric complex and has been implicated in the progression of ovarian, breast, hepatocellular carcinoma, bladder, and lung adenocarcinoma [19]. Although this gene has not been studied in an autophagy context using tumor samples, it was included in this study as it belongs to the autophagy network according to the Kumar et al. (2010) study [20]. Moreover, CENPK is overexpressed in cancers promoting proliferation through the PI3K-AKT signaling pathway, a pathway with a key regulatory role in autophagy [21].TUBB3 (Tubulin β 3 Class III) is associated with increased chemoresistance and poor prognosis in several cancers, including NSCLC, ovarian cancer, gastric cancer, breast cancer, uterine serous carcinoma, glioblastoma, colorectal cancer, and pancreatic ductal adenocarcinoma. It interacts with LC3, a key player in autophagosome formation [22].*TRIB3* (Tribbles Pseudokinase 3) overexpression inhibits the AKT-mTORC1 axis and autophagy-mediated cancer cell death [23]. *TRIB3* upregulation induced by ABTL0812, an anticancer agent under clinical development that induces *TRIB3* upregulation and potentiates common chemotherapy regimens in adenocarcinoma and squamous non-small cell lung cancer [24].MET is a receptor tyrosine kinase that activates the mTOR signaling pathway, regulating cell proliferation, apoptosis, autophagy, invasion, and tumorigenesis. The ubiquitination and degradation of MET can inhibit the proliferation, migration, and invasion of gastric cancer cells and induce apoptosis [25].TREM2 is a myeloid receptor expressed by tumor-infiltrating macrophages, commonly found within the tumor microenvironment of human cancers, and inversely correlated with prolonged survival in colorectal carcinoma and triple-negative breast cancer. TREM2 deficiency delays tumor growth in mice [26]. Moreover, it was observed that TREM2 regulates autophagy in tumor-associated microglia [27,28].

## 4. Clustering Solid Tumors Based on Autophagy-Related Genes

To assess the clustering of solid tumors, we utilized the expression levels of autophagy-related genes. To ensure robust observations, we focused on solid tumor types with more than 100 tumor and control samples. Employing this approach, we generated a UMAP plot that revealed three distinct clusters among the sixteen solid tumors analyzed (Figure 2A). The identified clusters were as follows: Cluster 0 comprised BRCA, KIRC, KIRP, LGG, KIHC, LUAD, LUSC, and PRAD; Cluster 1 included COAD, ESCA, PAAD, and STAD; and Cluster 2 consisted of GBM, OV, SKCM, and TGCT. Notably, cluster 0 grouped tissues with similar genetic or anatomical profiles, such as BRCA-PRAD, KIRC-KIRP, or LUAD-LUSC. Cluster 1 predominantly encompassed gastrointestinal tumors, while Cluster 2 included TGCT and OV, which are tumors from reproductive organs. To identify differentially expressed genes characterizing these clusters, we identified 18 genes that primarily distinguished Clusters 0 and 1. Cluster 2 exhibited decreased levels of these markers (Figure 2B), and therefore it will not be analyzed in further detail herein. Nevertheless, it is important to note that autophagy profiles have been induced and studied on SKCM, GBM, and OV models with anti-tumoral effects [29,30,31,32]. In addition, for GBM and SKCM, there is possible to suggest that expression similarities in non-pivotal genes could be originated at their division from the ectoderm, as was demonstrated for the *P2X7* receptor [33]. Then, these findings could support the evolutive hypothesis of cancer as an embryological phenomenon [34].

### 4.1. Autophagy Regulators Specific to Cluster 0

Our analysis revealed that the genes *ACTL6B*, *MAPT*, *PRKAA2*, *NUPR1*, *KRBA1*, *EEF1A2*, *TUBA3E*, and *TP53INP1* specifically characterized cluster “0” through their overexpression. Furthermore, we observed that these genes exhibited upregulated levels in tumors belonging to Cluster “0” compared to their normal adjacent tissues. While the limited number of normal-adjacent samples in the TCGA data introduces potential biases and limitations, the majority of the putative markers for Cluster “0” could be validated using the UALCAN tool [35]. A selection of these validation results is depicted in Figure 3.

#### 4.1.1. Protein and Mutational Features of Relevant Genes for Cluster 0

To gain insight into the protein products of these genes, we utilized the UALCAN tool to examine their change in proteins with data from the Clinical Proteomic Tumor Analysis Consortium (CPTAC) cohort. In the LIHC dataset, we found a comparable group of normal-adjacent samples, which demonstrated the upregulation of EEF1A2, NUPR1, and MAPT proteins in the tumor group (Figure 4). However, there were some inconsistencies. For instance, while the *PRKAA2* gene exhibited notable overexpression in LIHC, its corresponding protein was significantly downregulated (Appendix A). This disparity suggests the importance of considering the mutational profile of these genes or post-translational events on the produced proteins. Based on data from cBioPortal [36] of the TCGA datasets belonging to Cluster “0”, these genes exhibited a low frequency of mutations (1.1–5%, Figure 5), implying that somatic mutations may not play a significant role in dysregulating the relationship between autophagy-related genes and their corresponding proteins.

#### 4.1.2. Previous Research on Relevant Genes for Cluster 0

According to the MSigDB, the *NUPR1* and *PRKAA2* genes participate in macroautophagy and its regulation, while *MAPT* serves as an autophagy regulator, *TP53INP1* contributes to autophagosome organization, and *ACTL6B* and *KRBA1* are involved in the autophagy-related network. Additionally, other genes are associated with less-studied forms of autophagy. For example, *EEF1A2* is linked to chaperone-mediated autophagy and its regulation, *PRKAA2* is involved in lipophagy-related pathways, and *TUBA3E* is associated with aggrephagy.

Previous studies have highlighted the significant role of *NUPR1* in macroautophagy and its impact on the aggressiveness and treatment resistance of specific tumors such as BRCA, LUAD, LUSC, LIHC, and LGG [37,38,39,40,41,42,43]. *NUPR1*, also known as p8, is a transcriptional regulator that has been shown to reduce apoptosis caused by dihydroartemisinin (DHA), sorafenib, or ionizing radiation (IR) in LIHC tumor cells [37,38,39]. However, opposing effects have been observed in osteosarcoma and non-tumor cells [44]. Additionally, research has demonstrated that Δ9-tetrahydrocannabinol (THC) induces autophagy-mediated apoptosis in an LGG model [40]. Despite autophagy-related pathways being upregulated in LIHC and LGG tumors, it remains uncertain whether these pathways promote tumor growth or tumor suppression, necessitating further investigation. In lung and breast cancers (LUAD, LUSC, and BRCA), repression of *NUPR1*, in combination with conventional anticancer therapies, has been proven to control tumor growth [41,42,43]. Another study supports the inhibition of *NUPR1* using microRNA-637 (hsa-miR-637) as a promising option for this purpose [45].

Controversial results have emerged regarding the expression of the *PRKAA2* gene (Protein Kinase AMP-Activated Catalytic Subunit α 2, AMPKα2) in gastrointestinal malignancies [46,47,48,49]. Some studies suggest that repression of *PRKAA2* promotes tumor growth in gastrointestinal cancer by suppressing ferroptosis, an autophagy-dependent form of cell death [46]. On the other hand, other studies propose that *PRKAA2* activates autophagy-related pathways, leading to treatment resistance, and that its activation can be triggered by the gastrin hormone [47,48,49]. In LIHC, inhibiting *PRKAA2* has been shown to downregulate autophagy rates, and metformin has been identified as a potential *PRKAA* agonist for controlling hepatitis C virus (HCV) replication [50]. In glioma, low expression of *PRKAA2* has been associated with a better prognosis [51]. Although these findings may seem contradictory, especially considering that LGG belongs to Cluster “0”, they underscore the importance of including autophagy-related factors in the intra- and inter-individual heterogeneity of tumors.

The *MAPT* gene encodes the microtubule-associated protein tau, which has been extensively studied in Alzheimer’s disease (AD) [52,53]. Recent research has explored the interplay between autophagy and *MAPT* in AD and has demonstrated that overexpression of *MAPT*/tau inhibits the fusion of autophagosomes with lysosomes, leading to autophagosome accumulation through increased levels of LC3 protein [54]. Although direct links between *MAPT* and autophagy in cancer remain limited, the high expression levels of Tau protein in glioblastoma, a tumor with enhanced autophagy activity, have raised questions about its possible role in oncogenesis and its implications for cancer therapy [55].

In BRCA cohorts, a long non-coding RNA (lncRNA) for the *MAPT* gene called MAPT-AS1 has been found to be overexpressed in tumor tissues [56,57,58]. This is noteworthy because lncRNAs, which are usually located in antisense strands of DNA from original genes, can also be affected by somatic mutations irrespective of their canonical effects. Thus, the combination of somatic mutations and non-coding RNA as potential prognostic markers deserves further attention, as demonstrated in COAD [59].

*TP53INP1* gene exhibits inconsistent findings across experiments and tumor tissues. Some researchers have identified hsa-miR-106a as an oncomiR that targets *TP53INP1* in metastatic lung cancer [60], indicating its involvement in tumor suppression. Increasing the levels of *TP53INP1* could be crucial in controlling tumor growth through autophagy-dependent cell death. In the case of PRAD, hsa-miR-30a and hsa-miR-205 have been suggested as potential therapeutic options for suppressing *TP53INP1* [61,62]. However, it has been explained that *TP53INP1* is overexpressed as a response to ionizing radiation, which confers resistance [61,62]. Therefore, suppressing this gene could potentially resensitize tumor cells to standard treatment protocols. Like other representative genes in this cluster, *TP53INP1* exhibits a dual function. According to Peuget et al. (2021), oxidative stress induces the expression of *TP53INP1* [63]. This stress can trigger autophagy by interacting with LC3 in the cytoplasm or apoptosis by interacting with P53 in the nucleus, and the role of mitochondria and their metabolism in this process is also implicated [64]. Thus, an additional factor to consider in our analysis is the localization of autophagy-related transcripts and proteins. Unfortunately, there is insufficient information available to conduct this type of comparison.

In summary, NUPR1, PRKAA2, TP53INP1, ACTL6B, KRBA1, EEF1A2, and MAPT genes are coexpressed with 17 other genes (ANK2, ST8SIA1, GUCY2F, HERC1, TRHR, COL11A1, CHRM3, CNR2, KITLG, ROR1, CDKL5, PPOX, IGF2R, DDIT3, OPCML, ELOVL5, and BRINP2) according to the GeneMania database [65]. These genes are enriched in the MAPK pathway (*p* = 0.004) [66], allowing us to associate cluster “0” with a MAPK-dependent macroautophagy-like process. However, it is important to note the significant heterogeneity observed in the samples, classifications, tumor tissues, and other forms of autophagy. 

### 4.2. Tumors Balancing Macro- and Micro-Autophagy Processes (Clusters 0 and 1)

Clusters “0” and “1” in Figure 2 represent a distinct group of genes associated with tumors that exhibit a balance between macroautophagy and microautophagy processes. Notably, the genes *SREBF1*, *OPTN*, *ACBD5*, *SESN3*, *KERA*, *TUBA3D*, *FBXW7*, *TBC1D12*, *TLR9*, and *PLK2* show high expression levels in various tumors such as ESCA, PAAD, STAD, COAD, LUAD, LUSC, KIRC, LGG, PRAD, KIRP, LIHC, and BRCA. Furthermore, Figure 6 demonstrates that many of these genes are differentially expressed between tumor and normal-adjacent tissues.

In addition, after performing a random forest Gini importance analysis, we observed that KERA, TP53INP1, SREBF1, and TUBA3E showed great accuracy (above 75%) and over 75% of Gini contribution (Appendix A). It suggests the potential contribution of these autophagy-related genes in the classification of tissues regarding their dysregulation between tumor and normal samples.

Of particular interest are the *TUBA3D* and *FBXW7* genes, which are associated with the chaperone-mediated protein folding pathway (R-HSA-390466) according to the Enrichr database [66]. This suggests their potential involvement in chaperone-mediated autophagy. Supporting this idea, these genes have also been implicated in certain forms of microautophagy, such as aggrephagy and mitophagy, as indicated by the MSigDB (Appendix A). Additionally, these genes are part of the regulatory pathways of macroautophagy along with the other eight genes that cluster these tumors. *ACBD5*, *SREBF1*, and *OPTN* genes are also involved in microautophagy pathways, including aggrephagy, mitophagy, and xenophagy.

#### 4.2.1. Accumulation of ACBD5 Is Found in Tumors from Cluster 0 and 1

Notably, the *ACBD5* gene is interesting in autophagy-related studies as its deregulation can induce their accumulation at protein levels, as shown in Figure 7. This gene has been associated with peroxisome maintenance, lipid exchange, and homeostasis, which are crucial processes for lipid and carbohydrate metabolism reorganization in tumor cells [67,68]. These processes involve microautophagy pathways such as pexophagy, aggrephagy, and mitophagy [69].

#### 4.2.2. Previous Research on Overexpressed Genes in Tumors of Clusters 0 and 1 

Other genes related to microautophagy processes include *PLK2*, *SESN3*, *TLR9*, *OPTN*, and *SREBF1*. Independent research has demonstrated that the *PLK2* gene controls α-Synuclein aggregation in an autophagy-dependent context [70]. Although this process is dependent on macroautophagy and regulated by mTORC1 inhibition, it appears to be a microautophagy pathway that is specifically activated in the presence of its substrate, α-Synuclein [70,71]. An interesting regulatory axis involves the lncRNA OIP5-AS1, which targets hsa-miR-126 to prevent α-Synuclein aggregation in autophagy-activated cells [71].

Regarding the *SESN3* gene, recent studies have identified its role as an autophagy activator in tumor cells by repressing mTORC1 [72]. However, this gene has also been associated with other autophagy pathways such as chaperone-mediated autophagy [73]. Overexpression of *SESN3* has been observed in LUAD [73] and ESCA [74] models, suggesting its potential involvement in promoting pro-tumor autophagy pathways. Expression levels of this gene can be regulated by specific miRNAs, such as hsa-miR-194-3p [73] or hsa-miR-429 [74].

About mitophagy, several reports have described the upregulation of the *TLR9* gene in tumors belonging to Clusters “0” and “1” [75,76,77], indicating its involvement in inducing this form of autophagy. In BRCA, it has been reported that this gene plays a role in the rewiring of doxorubicin and may explain the cardiomyocyte death and systolic dysfunction observed in patients undergoing this tumor treatment [78]. Consistent with these findings, *TLR9* was found to be upregulated in aggressive versions of LIHC, LUAD, LUSC, and COAD models [79,80,81]. Consequently, various regulatory pathways have been proposed to control *TLR9* expression. For example, hsa-miR-30a has been shown to sensitize LIHC cells to a combined therapy of hydroxychloroquine and sorafenib by repressing *TLR9* [79]. On the other hand, inducing *TLR9* expression in dendritic cells has been suggested as a potential therapeutic strategy, as demonstrated in PAAD cases [82]. It is important to note that bulk analyses using next-generation sequencing (NGS) do not differentiate between the origins of cells within tumors, which can lead to different interpretations of the results. Therefore, researchers are increasingly turning to single-cell sequencing to differentiate immune cells, tumor cells, and normal-adjacent cells with varying autophagy-related profiles within the same tumor pool.

In addition to *TLR9*, *OPTN* has been extensively studied in the context of mitophagy. *PINK1* and *PRKN*, which are highly studied autophagy-related genes, are also involved in this process. The *PHB2* gene stabilizes *PINK1* in mitochondria, facilitating the recruitment of Parkin (the product of *PRKN*), ubiquitin, and optineurin (the product of *OPTN*) to promote mitophagy [83,84,85,86]. However, a recent study challenges the necessity of *PINK1* and *PRKN* for initiating mitophagy [87]. Consequently, it has been suggested that *OPTN* may have tumor suppressor functions by activating suppressor autophagy mediated by *HACE1*, a tumor suppressor [88,89,90], or by repressing the pro-oncogenic transforming growth factor-β (TGFβ) signaling in triple-negative breast cancer (TNBC) cells, a subtype of BRCA [91]. Importantly, *OPTN* has been found to be downregulated in GBM tumor samples, which has been corroborated by independent studies [92]. The same study proposes that inducing *OPTN* expression in GBM cells could help control tumor growth, supporting a suppressive role for this gene, although the underlying mechanism remains unknown.

In terms of the application of *OPTN* in the context of mitophagy and the tumor environment, several studies have identified *OPTN* as a potential therapeutic target. For instance, it has been observed that *OPTN* induces pro-tumor mitochondrial-related autophagy, reducing the efficacy of combined treatments involving pemetrexed, cisplatin, and MEK inhibitors or anti-PD-L1 in a LUSC model [80]. In a PAAD model, repression of *OPTN* leads to apoptosis through chaperone-mediated autophagy [93]. Similar to *TLR9*, understanding the function of *OPTN* allows us to differentiate its contribution to tumor growth based on its expression in surrounding cells. In LUAD models, higher expression of *OPTN* in fibroblasts surrounding the tumor contributes to tumor invasiveness [94].

*SREBF1* upregulation has been linked to mTORC1-dependent autophagy, which may be induced by leptins to suppress ferroptosis in BRCA, LIHC, PRAD, and LUAD models [95,96,97,98]. Additionally, *SREBF1* levels were found to be elevated in PAAD tissues, regulated by high glucose concentrations. In PAAD models, the upregulation of *SREBF1* helps control autophagy levels [99]. This gene may act as a negative regulator of mTORC1-dependent autophagy, favoring pro-tumor microautophagy pathways. It is worth noting that *SREBF1* can function as both a protein and a transcription factor. Studies have demonstrated that genes upregulated by the *SREBF1* transcription factor can be altered in the presence of cisplatin, inducing treatment resistance in a LUSC model [100]. This evidence highlights the importance of carefully analyzing autophagy-related genes with dual functions to enhance our understanding of this process. A study proposed that mTORC2 stabilizes *SREBF1* through *FBXW7*-mediated regulation to integrate autophagy and lipid metabolism processes, leading to the downregulation of target genes such as acetyl-CoA carboxylase and fatty-acid synthase [101].

Considering the combined findings of two genes involved in tumor clusterization, *FBXW7* and *SREBF1*, it is hypothesized that these tumors exchange autophagy-related processes and large-scale technologies based on their aggressiveness and treatment sensitivity or resistance. However, conducting large-scale high-throughput analyses in mass groups could obscure specific autophagy pathways in certain tumor subtypes or patients. Therefore, the current perspective is to compare global observations with focused research. Nevertheless, the scientific community is moving towards a comprehensive analysis of tumors, considering their heterogeneity and subclonal profile, which will allow us to confirm our current hypotheses about autophagy-related processes in the tumor environment in the future.

Regarding macroautophagy, the *FBXW7* gene has been the focus of numerous studies aiming to characterize its function. This gene is known as a tumor suppressor as it is frequently mutated or suppressed in human tumors [102]. However, its dysregulation in chemoresistance remains controversial, suggesting that its behavior depends on the context. It has been observed to be upregulated in resistant gastric cells [103] and downregulated in chemoresistant models of BRCA [104].

Interestingly, *FBXW7* has been found to induce the expression of *ATG16L1*, an important gene involved in LC3 lipidation and autophagosome formation, while not affecting the levels of other autophagy-related genes (ATG) [105]. Moreover, *FBXW7* suppresses mTORC1, thereby activating autophagy pathways [106,107]. *FBXW7* participates in different molecular axes, resulting in different effects on tumor cells. For instance, the GSK3-FBXW7 interaction leads to the ubiquitination and degradation of Rictor, increasing cellular ROS (reactive oxygen species) in an autophagy-activated context [108]. On the other hand, interactions between *FBXW7* and oncogenes such as *SHOC2* or *LSD1* can reduce the expression of autophagy-related pathways [106,107,109]. In conditions where tumors are growing, cisplatin treatments have been shown to induce the degradation of the MRE11-RAD50-NBS1 (MRN) complex by *FBXW7* and lysosomes [102]. As a result, the overexpression of the MRN complex or the suppression of the *FBXW7* gene can lead to cisplatin-resistant tumors and a poor prognosis. In relation to this, hsa-miR-25 and hsa-miR-223 have been shown to suppress *FBXW7* levels, promoting autophagy and treatment resistance in LIHC [110] and LUAD [111] models, respectively. Anti-miRs could be used to counteract the suppression of *FBXW7* levels, but it is important to better understand the specific context in which this strategy would be applicable.

Lastly, three genes (*TBC1D12*, *KERA*, and *TUBA3D*) that contribute to tumor clustering in groups “0” and “1” have not been previously associated with the tumor-related autophagy process. It is important to emphasize that, in our analysis, the *KERA* gene was the top gene in Gini relevance and accuracy in tissue pooling of groups between 0 + 1 vs. 2. Studies on mutations in the *TBC1D12* gene (TBC1 Domain Family Member 12) have been conducted in urological tumors, suggesting that alterations in its mutational profile could be linked to worse patient survival [112]. Interestingly, this gene exhibits a higher mutation frequency in PRAD samples compared to other patients (Figure 8). The *KERA* (Keratan Sulfate Proteoglycan Keratocan) gene has been found to have lower levels in cisplatin and paclitaxel-resistant OV models [113], partially aligning with observations in the entire dataset (Cluster “2”). The expression levels of the *TUBA3D* (Tubulin α-3D Chain) gene in BRCA (upregulated) and OV (downregulated) have been validated [114,115]. Notably, in BRCA models, *TUBA3D* was shown to be downregulated in paclitaxel-resistant cells compared to parental cells [116].

In summary, the findings presented in this discussion suggest that all the aforementioned genes may make significant contributions to tumor-related autophagy through their expression in tumors and the surrounding cells, warranting further attention in future research.

## 5. Conclusions

The availability of large cancer datasets has provided an extensive evidence-based approach to understanding the role of autophagy-related genes in various human cancers and their clinical implications, including cancer progression, development, and treatment response. In this study, we utilized different databases to analyze the expression levels of these genes and their associations. Through our analyses, we identified commonly overexpressed genes across the three approaches while also recognizing specific genes in each analysis.

Furthermore, by examining the expression patterns of autophagy-related genes, we were able to classify the 16 solid tumors into 3 distinct clusters. Clusters 0 and 1 exhibited significant involvement of key autophagy-related genes, suggesting shared metabolic pathways and potentially similar therapeutic responses related to autophagy within each tumor type. Interestingly, we also discovered three genes (*TBC1D12*, *KERA*, and *TUBA3D*) that have not been previously associated with tumor-related autophagy.

The comprehensive analysis of autophagy-related clusters in solid tumors, combined with real-world data, holds great potential for identifying therapy targets and conducting further mechanistic studies. However, it is important to acknowledge the limitations of our study. Primarily, our analysis was based on gene expression data, lacking the ability to differentiate between the various cell types within the tumor microenvironment and lacking spatial information about specific molecules. Therefore, additional research and experimental validation are necessary to explore the potential significance of these genes in cancer development and treatment.

## Figures and Tables

**Figure 1 genes-14-01550-f001:**
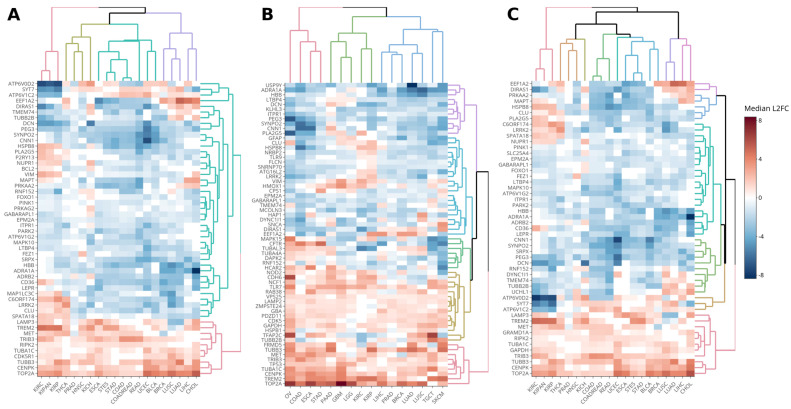
Autophagy-related genes have differentiated expressions in solid tumors. Heatmaps showing the following values: (**A**) Log_2_ fold-change (L2FC) in the medians of expression levels from normal to tumor tissues among TCGA participants; (**B**) L2FC in the medians of expression levels from normal to tumor tissues among TCGA + GTEX + TARGET participants; (**C**) medians of normal to tumor tissues L2FC matched by TCGA participants. White cells represent genes without statistical differences between tumor and normal (or normal-adjacent) tissues. The statistical test applied were Mann–Whitney’s test (**A**,**B**) and Wilcoxon’s test (**C**). BLCA: bladder urothelial carcinoma; BRCA: breast invasive carcinoma; CHOL: cholangiocarcinoma; COAD: colon adenocarcinoma; ESCA: esophageal carcinoma; GBM: glioblastoma multiforme; HNSC: head and neck squamous cell carcinoma; KICH: kidney chromophobe; KIRC: kidney renal clear cell carcinoma; KIRP: kidney renal papillary cell carcinoma; LGG: brain lower grade glioma; LIHC: liver hepatocellular carcinoma; LUAD: lung adenocarcinoma; LUSC: lung squamous cell carcinoma; OV: ovarian serous cystadenocarcinoma; PRAD: prostate adenocarcinoma; READ: rectum adenocarcinoma; SKCM: skin cutaneous melanoma; STAD: stomach adenocarcinoma; TGCT: testicular germ cell tumors; THCA: thyroid carcinoma; UCEC: uterine corpus endometrial carcinoma; COADREAD: colorectal adenocarcinoma (COAD + READ); KIPAN: pan-kidney cohort (KICH + KIRC + KIRP); STES: stomach and esophageal carcinoma (STAD + ESCA).

**Figure 2 genes-14-01550-f002:**
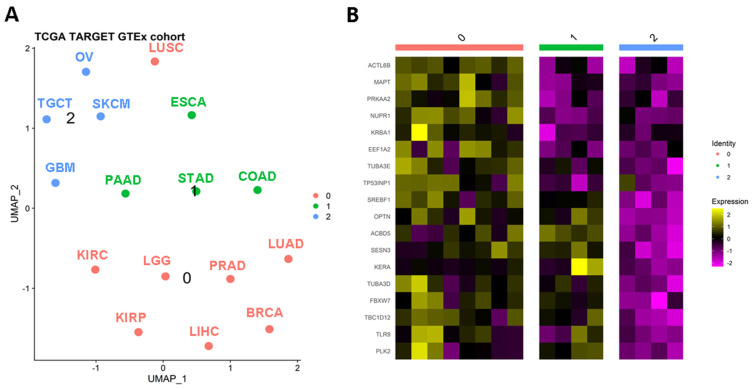
Autophagy-related genes can stratify solid tumors. (**A**) Clusterization of solid tumors based on the differential expression of autophagy genes. After a UMAP analysis, it is possible to recognize three classifications (**B**) of relevant tumors based on the expression of autophagy genes. BRCA: breast invasive carcinoma; COAD: colon adenocarcinoma; ESCA: esophageal carcinoma; GBM: glioblastoma multiforme; KIRC: kidney renal clear cell carcinoma; KIRP: kidney renal papillary cell carcinoma; LGG: brain lower grade glioma; LIHC: liver hepatocellular carcinoma; LUAD: lung adenocarcinoma; LUSC: lung squamous cell carcinoma; OV: ovarian serous cystadenocarcinoma; PAAD: pancreatic adenocarcinoma; PRAD: prostate adenocarcinoma; SKCM: skin cutaneous melanoma; STAD: stomach adenocarcinoma; TGCT: testicular germ cell tumors.

**Figure 3 genes-14-01550-f003:**
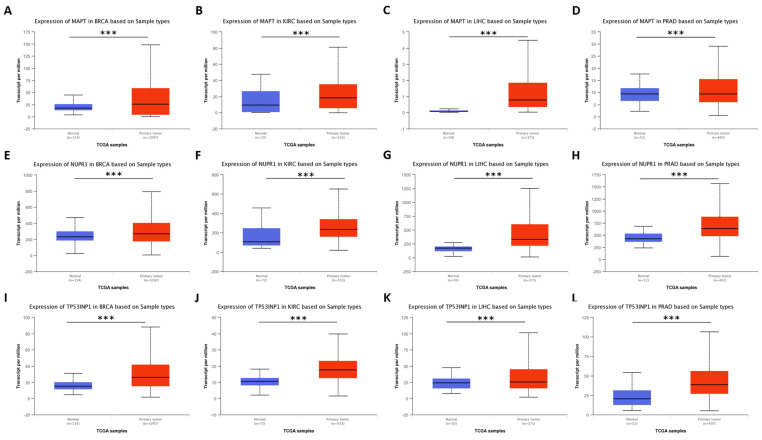
Genes upregulated in Cluster “0” differentiate tumor and normal adjacent tissues. Using the UALCAN tool, we compared a selection of genes stratifying solid tumors in Cluster “0” between tumor and normal-adjacent tissues. Herein, we represent data for BRCA, KIRC, LIHC, and PRAD datasets for the *MAPT* (**A**–**D**), *NUPR1* (**E**–**H**), and *TP53INP1* (**I**–**L**) genes. *** represents comparisons with *p*-value < 0.001 on Welch’s *t*-test. BRCA: breast invasive carcinoma; KIRC: kidney renal clear cell carcinoma; LIHC: liver hepatocellular carcinoma; PRAD: prostate adenocarcinoma.

**Figure 4 genes-14-01550-f004:**
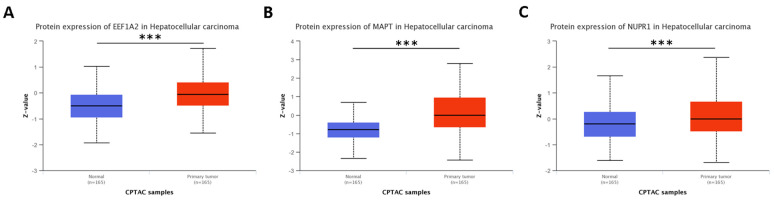
Putative autophagy-related gene markers codify dysregulated proteins in LIHC. Data from Clinical Proteomic Tumor Analysis Consortium (CPTAC) and the International Cancer Proteogenome Consortium (ICPC) datasets via the UALCAN tool allow us to confirm putative gene markers upregulated in LIHC with their proteic version upregulated. Here is the shown data for EEF1A2 (**A**), MAPT (**B**), and NUPR1 (**C**) proteins. *** represents comparisons with *p*-value < 0.001 on Welch’s *t*-test. LIHC: liver hepatocellular carcinoma.

**Figure 5 genes-14-01550-f005:**
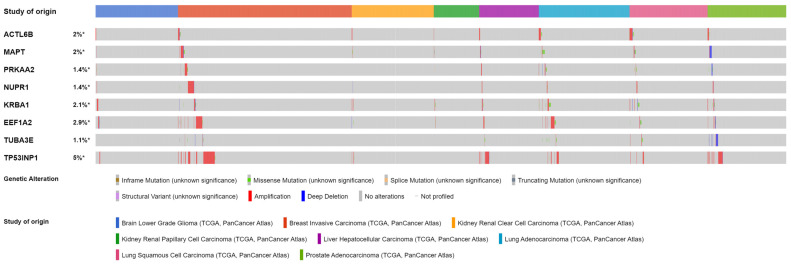
Putative autophagy-related gene markers are rarely mutated in solid tumors of Cluster “0”. Oncoprint produced by the cBioPortal for Cancer Genomics shows the frequency of somatic mutations per gene and cancer dataset related to Cluster “0”. Notably, a group of BRCA patients showed amplifications of all genes, whereas some PRAD patients showed deletions in *MAPT*, *PRKAA2*, and *TUBA3E* genes. * means that the mutational frequency was estimated about the number of profiled patients as this number can vary between genes.

**Figure 6 genes-14-01550-f006:**
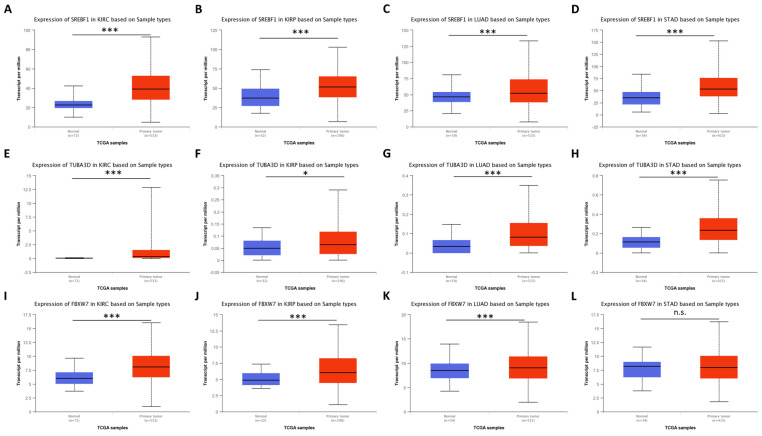
Genes upregulated in Clusters “0” and “1” differentiate tumor and normal adjacent tissues. Using the UALCAN tool, we compared a selection of genes stratifying solid tumors in Clusters “0” and “1” between tumor and normal-adjacent tissues. Herein, we represent data for KIRC, KIRP, LUAD, and STAD datasets for the *SREBF1* (**A**–**D**), *TUBA3D* (**E**–**H**), and *FBXW7* (**I**–**L**) genes. *p*-values on Welch’s *t*-test are shown as *** (*p* < 0.001); * (*p* < 0.05); n.s. (*p* ≥ 0.05). KIRC: kidney renal clear cell carcinoma; KIRP: kidney renal papillary cell carcinoma; LUAD: lung adenocarcinoma; STAD: stomach adenocarcinoma.

**Figure 7 genes-14-01550-f007:**
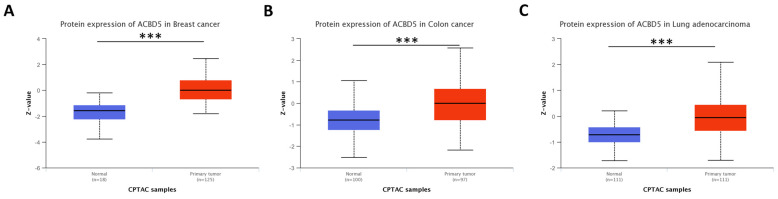
ACBD5 protein levels in tumors belonging to Clusters “0” and “1”. Data from Clinical Proteomic Tumor Analysis Consortium (CPTAC) and the International Cancer Proteogenome Consortium (ICPC) datasets via the UALCAN tool allow us to confirm dysregulated levels of the ACBD5 protein in three tumor tissues (compared with their respective non-tumor adjacent tissues). Here is the shown data for BRCA (**A**), COAD (**B**), and LUAD (**C**) datasets. *** represents comparisons with *p*-value < 0.001 on Welch’s *t*-test. BRCA: breast invasive carcinoma; COAD: colon adenocarcinoma; LUAD: lung adenocarcinoma.

**Figure 8 genes-14-01550-f008:**
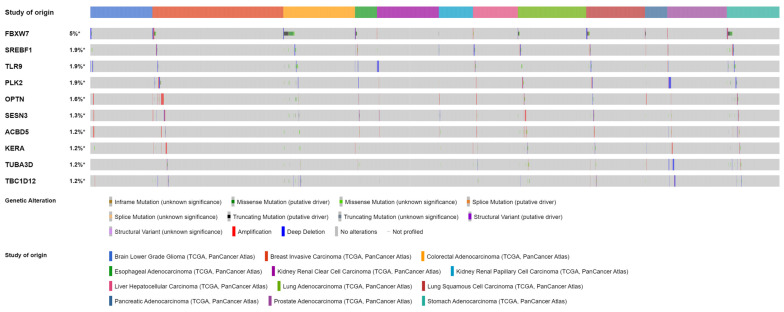
Putative autophagy-related gene markers are rarely mutated in solid tumors of clusters “0” and “1”. Oncoprint produced by the cBioPortal for Cancer Genomics shows the frequency of somatic mutations per gene and cancer dataset related to the clusters “0” and “1”. Notably, the *FBXW7* accounts for the higher mutational frequency, mainly in the COAD dataset, whereas the PRAD cohort shows a high percentage of patients with deleted regions of analyzed genes. * means that the mutational frequency was estimated about the number of profiled patients as this number can vary between genes.

## Data Availability

No new data were created or analyzed in this study. Data sharing does not apply to this article.

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
