# Peer review of "Insights from a Computational-Based Approach for Analyzing Autophagy Genes across Human Cancers"

_genes, 2023, doi:10.3390/genes14081550_

Round 1

Reviewer 1 Report

Summary: Autophagy plays a role in cell biology, and it has become the subject of studies in cancer through the years. The relevance of autophagy-related genes in cancer, and the implications as prognostic markers, provide new insights for cancer therapies.

The review provides relevant insights regarding the autophagy-related genes in cancer (TBC1D12, KERA, and TUBA3D). As mentioned before, the topic is relevant and has gained attention to overcome tumor resistance to available treatments. A weak point of the manuscript worth mentioning is the literature used to specify autophagy genes. In 2022, a Pan-cancer gene expression analysis used robust literature to specify autophagy genes in the analysis. This article was not referred to in this review (please see reference below). It is possible that if more autophagy-related genes were used in the analysis, that could have helped to better define clusters and to reveal potential new autophagy-related genes in cancer.

((Pan-cancer gene expression analysis: Identification of deregulated autophagy genes and drugs to target them. Accepted 12 August 2022. https://doi.org/10.1016/j.gene.2022.146821))

There are a few unclear details in the text regarding the way it was structured. Even though it is appreciating the attempt to make the text concise, a few references were lacking, as pointed out in the specific lines (please see specific comments). Also, another point worth mentioning is regarding the concept of validation, mentioned in the text (please see specific comments).

Specific comments

97 – Because they used Molecular Signature Database (MSigDB) from 2015 (ref 9), it is possible other genes related to autophagy were carried out of the analysis. It would be important to consider that in their conclusion, as of an updated literature regarding genes associated with autophagy.

141 - This gene encodes a DNA topoisomerase, an enzyme that controls and alters the topologic states of DNA during transcription. This nuclear enzyme is involved in processes such as chromosome condensation, chromatid separation, and the relief of torsional stress that occurs during DNA transcription and replication. It is not involved in DNA metabolism, therefore a more appropriate term could be used on the referred phrase to avoid confusion.

146 - CENPK, a cell cycle-related gene, contributes to malignant progression and prognosis. CENPK contributed to cell proliferation, colony formation, and tumorigenesis of castration-resistant prostate cancer, for example. Because the topic is autophagy-related genes, it would be important to bring a reference for each described gene to support the inclusion of these genes in the topic as autophagy-related gene.

178 – According to the author’s analysis, cluster 2 also includes GBM and SKCM. Is there any reason it was decided to mention the other half only?

191 – The following mentioned genes are not all in Fig 1, for example (KRBA1, ACTL6B, TUBA3E, TP53INP1). Are they not differentially expressed in comparison to normal tissue but only between the different clusters?

197 – What would make UALCAN more powerful than TCGA regarding normal tissue, as brought up in line 196? Also, Figure 3, UALCAN shows results based on transcription level, therefore same concept as gene expression from TCGA accessed by Xena. It creates a level of confusion if another gene expression platform is used for validation. Instead, to validate if a gene is differentially expressed or not, it is common to check for the protein levels instead. Or perhaps the authors could elaborate more about validation on this case.

201 – Would it be EEF1A2, instead of EEF1A1?

205 – They could also consider post-translational regulation to explain decrease protein levels. It is not uncommon to increase mRNA to compensate for post-translational regulation of proteins.

226 – There is an error on the legend.

243 – The authors could include references to the first phrase.

267 – Also, to add reference on the first phrase as well.

274 - PRKAA2 is not present in Figure 1B regarding GBM. Would it be because it has the same level as normal tissue? But the suggested literature shows that it could be relevant. However, in the present manuscript this gene was not differentially expressed (higher or lower) vs control tissue.

282 – It seems from the text (line 201), it was written EEF1A1. But in the figure, it is A2.

283 – Not only here but all the statistics applied, it would be better to include the type or name of statistics in the legend of the figure.

308 – What would be the reference that supports that line regarding TP53INP1?

339 – The number of the topic is the same as the previous one.

345- On Fig2, these genes were used to distinguish and cluster tumors. But some of these genes were not differentially expressed vs normal tissue on Fig 1. But Fig 6 shows that as a transcription per million. Could the authors explain the difference between TCGA accessed by Xena and the UALCAN in matters of validation, if both are depicted as transcription but not protein translation?

482 – The legend seems to be misplaced from the figure.

Author Response

Reviewer #1

Point 1.1:

Summary: Autophagy plays a role in cell biology, and it has become the subject of studies in cancer through the years. The relevance of autophagy-related genes in cancer, and the implications as prognostic markers, provide new insights for cancer therapies.

The review provides relevant insights regarding the autophagy-related genes in cancer (TBC1D12, KERA, and TUBA3D). As mentioned before, the topic is relevant and has gained attention to overcome tumor resistance to available treatments. 

Response 1.1:

Thank you for this summary of our study and for emphasizing the relevance of analyzing autophagy genes in cancer.

Point 1.2:

A weak point of the manuscript worth mentioning is the literature used to specify autophagy genes. In 2022, a Pan-cancer gene expression analysis used robust literature to specify autophagy genes in the analysis. This article was not referred to in this review (please see reference below). It is possible that if more autophagy-related genes were used in the analysis, that could have helped to better define clusters and to reveal potential new autophagy-related genes in cancer.

((Pan-cancer gene expression analysis: Identification of deregulated autophagy genes and drugs to target them. Accepted 12 August 2022. https://doi.org/10.1016/j.gene.2022.146821))

Response 1.2:

Thank you for pointing out this limitation. In the cited article, the authors used a previous version of the Molecular Signature Database (MSigDB) and the Human Autophagy Database (HADb) to get 574 autophagy-related genes. In our study, we used an updated version of MSigDB, focusing on curated (C2) and Oncology-related (C5) datasets retrieving 707 genes. After comparing both lists, we have 440 in-common genes, representing 76.7% of the studied genes in that paper and 62.2% of the genes analyzed in our study.

As our knowledge about autophagy in cancer is still developing evidence, we aim to propose analytical insights to add novel genes to bench experiments. Nevertheless, these analyses are sensitive to improvements as more robust and curated information will be available. In this version, we re-organized the topics to clarify the methods and rationale followed to run our study. We included a new Figure S1 to support our rationale and complement the description on lines 94-131. Herein, we have cited the proposed reference, in addition to studies supporting HADb and the Human Autophagy Modulator Database (HAMdb).

Point 1.3:

There are a few unclear details in the text regarding the way it was structured. Even though it is appreciating the attempt to make the text concise, a few references were lacking, as pointed out in the specific lines (please see specific comments). Also, another point worth mentioning is regarding the concept of validation, mentioned in the text (please see specific comments).

Response 1.3:

Thank you for the specific observations about our study. We have revised the text accordingly with your comments. We believe it has been improved. Please, take a look at each specific statement.

Point 1.4:

97 – Because they used Molecular Signature Database (MSigDB) from 2015 (ref 9), it is possible other genes related to autophagy were carried out of the analysis. It would be important to consider that in their conclusion, as of an updated literature regarding genes associated with autophagy.

Response 1.4:

Thank for your comment. In our study, we used an updated version of MSigDB (released on March 2023), focusing on curated (C2) and Oncology-related (C5) datasets retrieving 707 genes.  As our knowledge about autophagy in cancer is still developing evidence, we aim to propose analytical insights to add novel genes to bench experiments. Nevertheless, these analyses are sensitive to improvements as more robust and curated information will be available. In this version, we re-organized the topics to clarify the methods and rationale followed to run our study. We included a new Figure S1 to support our rationale and complement the description on lines 94-131.

Point 1.5:

141 - This gene encodes a DNA topoisomerase, an enzyme that controls and alters the topologic states of DNA during transcription. This nuclear enzyme is involved in processes such as chromosome condensation, chromatid separation, and the relief of torsional stress that occurs during DNA transcription and replication. It is not involved in DNA metabolism, therefore a more appropriate term could be used on the referred phrase to avoid confusion.

Response 1.5:

We apologize for having used an incorrect term. In this version, we changed “DNA metabolism” to “DNA topology rearrangements” (lines 148-149). We believe this change is consistent with the function of the enzyme TOP2A.

Point 1.6:

146 - CENPK, a cell cycle-related gene, contributes to malignant progression and prognosis. CENPK contributed to cell proliferation, colony formation, and tumorigenesis of castration-resistant prostate cancer, for example. Because the topic is autophagy-related genes, it would be important to bring a reference for each described gene to support the inclusion of these genes in the topic as autophagy-related gene.

Response 1.6:

Thank you for having commented on this topic. To the best of our knowledge, there is no previous literature analyzing CENPK in the autophagy context of tumors. Nevertheless, this gene was studied as it belongs to an autophagy network proposed by Kumar et al. (2010, ref 20). We have complemented this on lines 155-159 and also discuss another study (Wu et al., 2021, ref 21) that could explain the association of CENPK and autophagy.

Point 1.7:

178 – According to the author’s analysis, cluster 2 also includes GBM and SKCM. Is there any reason it was decided to mention the other half only?

Response 1.7:

Thank you for pointing out this. We have described all tumors belonging to cluster 2 in lines 208-209: “Cluster 2 consisted of GBM, OV, SKCM, and TGCT”. After this description, we commented on possible similarities between these tumors. Thus, OV and TGCT are tumors from reproductive organs (lines 211-212), while GBM and SKCM are generated from the ectoderm (lines 217-220).

Point 1.8:

191 – The following mentioned genes are not all in Fig 1, for example (KRBA1, ACTL6B, TUBA3E, TP53INP1). Are they not differentially expressed in comparison to normal tissue but only between the different clusters?

Response 1.8:

We apologize if our workflow was not clearly presented. Figure 1 shows relevant autophagy-related genes that are differentially expressed between tumor and normal samples following different analytical strategies. Due to the large number of genes in analysis (707), this figure shows genes differentially expressed in many tissues. Then, genes specifically dysregulated in less than eight tumors are not shown.  Nevertheless, these genes are key to differentiate tissue clusters in Figure 2. Therefore, there is possible that many genes representing clusters in Figure 2 are not shown in Figure 1.

Point 1.9:

197 – What would make UALCAN more powerful than TCGA regarding normal tissue, as brought up in line 196? Also, Figure 3, UALCAN shows results based on transcription level, therefore same concept as gene expression from TCGA accessed by Xena. It creates a level of confusion if another gene expression platform is used for validation. Instead, to validate if a gene is differentially expressed or not, it is common to check for the protein levels instead. Or perhaps the authors could elaborate more about validation on this case.

Response 1.9:

We used UALCAN to evaluate gene and protein levels from TCGA and CPTAC patients. We used this webtool as a reproducible and previously published technique to verify our analyses performed using FirebrowseR for same data (TCGA) and Xena repository to retrieve expression levels from the TCGA + GTEx + TARGET study. We apologize if our workflow was not clearly shown before. In this version, we included a new Figure S1 to support our rationale and complement the description on lines 94-131.

Point 1.10:

201 – Would it be EEF1A2, instead of EEF1A1?

Response 1.10:

Sorry for this mistake. We have corrected it on line 262.

Point 1.11:

205 – They could also consider post-translational regulation to explain decrease protein levels. It is not uncommon to increase mRNA to compensate for post-translational regulation of proteins.

Response 1.11:

Thank you for pointing out this. We agree with your comment. We have changed this statement to be “This disparity suggests the importance of considering the mutational profile of these genes or post-translational events on the produced proteins.” (lines 239-241).

Point 1.12:

226 – There is an error on the legend.

Response 1.12:

We apologize for this mistake. We have improved the structure of the document, including figures and their legends.

Point 1.13:

243 – The authors could include references to the first phrase.

Response 1.13:

Thank you for the comment. We have added the proper references in line 267.

Point 1.14:

267 – Also, to add reference on the first phrase as well.

Response 1.14:

Thank you for the comment. We have added the proper references in line 290.

Point 1.15:

274 - PRKAA2 is not present in Figure 1B regarding GBM. Would it be because it has the same level as normal tissue? But the suggested literature shows that it could be relevant. However, in the present manuscript this gene was not differentially expressed (higher or lower) vs control tissue.

Response 1.15:

We apologize if our workflow was not clearly presented. Figure 1 shows relevant autophagy-related genes that are differentially expressed between tumor and normal samples following different analytical strategies. Due to the large number of genes in analysis (707), this figure shows genes differentially expressed in many tissues. Then, genes specifically dysregulated in less than eight tumors are not shown.  Nevertheless, these genes are key to differentiate tissue clusters in Figure 2. Therefore, there is possible that many genes representing clusters in Figure 2 are not shown in Figure 1.

Point 1.16:

282 – It seems from the text (line 201), it was written EEF1A1. But in the figure, it is A2.

Response 1.16:

Sorry for this mistake. We have corrected it on line 262.

Point 1.17:

283 – Not only here but all the statistics applied, it would be better to include the type or name of statistics in the legend of the figure.

Response 1.17:

Thank you for commenting on this. We have added all proper statistical descriptions in the legend of figures.

Point 1.18:

308 – What would be the reference that supports that line regarding TP53INP1?

Response 1.18:

Thank you for the comment. We have added the proper references in line 331.

Point 1.19:

339 – The number of the topic is the same as the previous one.

Response 1.19:

We apologize for this mistake. We have corrected the error and reorganized the structure of our review paper.

Point 1.20:

345- On Fig2, these genes were used to distinguish and cluster tumors. But some of these genes were not differentially expressed vs normal tissue on Fig 1. But Fig 6 shows that as a transcription per million. Could the authors explain the difference between TCGA accessed by Xena and the UALCAN in matters of validation, if both are depicted as transcription but not protein translation?

Response 1.20:

We used UALCAN to evaluate gene and protein levels from TCGA and CPTAC patients. We used this webtool as a reproducible and previously published technique to verify our analyses performed using FirebrowseR for same data (TCGA) and Xena repository to retrieve expression levels from the TCGA + GTEx + TARGET study. We apologize if our workflow was not clearly shown before. In this version, we included a new Figure S1 to support our rationale and complement the description on lines 94-131.

Point 1.21:

482 – The legend seems to be misplaced from the figure.

Response 1.21:

We apologize for this mistake. We have improved the structure of the document, including figures and their legends.

Reviewer 2 Report

The review manuscript focus on autophagy-related genes in different tumors. Authors have performed gene expression analyses to understand gene regulation and identify autophagy-related clusters. Further gene-level characterization of these clusters is comprehensively discussed. In general, the content of the manuscript is fine but needs structural improvements. I have the following concerns-

Line 489: Have you analyzed the contributions of these three genes TBC1D12, KERA, and TUBA3D to tumor clustering? It is important to discuss their contribution.

Both subheadings have the same numbers: “2.2.1. Autophagy regulators specific to cluster 0” and “2.2.1. Tumors balancing macro- and micro-autophagy processes (clusters 0 and 1)”

The discussion on autophagy-related genes in cluster 0 (section 2.2.1) seems to contain everything in one section. I would strongly recommend partitioning this whole section. For example, the discussion on mutation along with Figure 5 can be put with a separate sub-heading. Similarly, the text with a discussion on protein expression validation should be separated with proper heading. I would suggest dividing the discussion text into sections with proper headings. Similar suggestion for “2.2.1. Tumors balancing macro- and micro-autophagy processes (clusters 0 and 1)”.

This should go to a different section: “2.1. Autophagy-related Genes differentially expressed in solid tumors”

As this is a review article, please include a systematic discussion on data collection, methods included, articles reviewed etc.

Page 14: The Figure on this page doesn’t have a number. 

Author Response

Reviewer #2

Point 2.1:

The review manuscript focus on autophagy-related genes in different tumors. Authors have performed gene expression analyses to understand gene regulation and identify autophagy-related clusters. Further gene-level characterization of these clusters is comprehensively discussed. In general, the content of the manuscript is fine but needs structural improvements. I have the following concerns

Response 2.1:

Thank you for this summary of our study and for emphasizing the relevance of analyzing autophagy genes in cancer. We have revised the text accordingly with your comments. We believe it has been improved. Please, take a look at each specific statement.

Point 2.2:

Line 489: Have you analyzed the contributions of these three genes TBC1D12, KERA, and TUBA3D to tumor clustering? It is important to discuss their contribution.

Response 2.2:

We apologize if our workflow was not clearly presented. Figure 1 shows relevant autophagy-related genes that are differentially expressed between tumor and normal samples following different analytical strategies. Due to the large number of genes in analysis (707), this figure shows genes differentially expressed in many tissues. Then, genes specifically dysregulated in less than eight tumors are not shown.  Nevertheless, these genes are key to differentiate tissue clusters in Figure 2. Therefore, there is possible that many genes representing clusters in Figure 2 are not shown in Figure 1.

Then, TBC1D12, KERA, and TUBA3D contribute to the clusterization of tumors as shown in Figure 2B, in combination with other genes. Nevertheless, the relevance of these three genes is focused on the lack of studies in an autophagy context using tumor samples. That is the reason to suggest TBC1D12, KERA, and TUBA3D as candidates for further experiments in tumors of clusters 0 and 1.

Point 2.3:

Both subheadings have the same numbers: “2.2.1. Autophagy regulators specific to cluster 0” and “2.2.1. Tumors balancing macro- and micro-autophagy processes (clusters 0 and 1)”

Response 2.3:

We apologize for this mistake. We have corrected the error.

Point 2.4:

The discussion on autophagy-related genes in cluster 0 (section 2.2.1) seems to contain everything in one section. I would strongly recommend partitioning this whole section. For example, the discussion on mutation along with Figure 5 can be put with a separate sub-heading. Similarly, the text with a discussion on protein expression validation should be separated with proper heading. I would suggest dividing the discussion text into sections with proper headings. Similar suggestion for “2.2.1. Tumors balancing macro- and micro-autophagy processes (clusters 0 and 1)”.

Response 2.4:

Thank you for this comment. We have and reorganized the structure of our review paper.

Point 2.5:

This should go to a different section: “2.1. Autophagy-related Genes differentially expressed in solid tumors”

Response 2.5:

Thank you for pointing out this. We have corrected the error and reorganized the structure of our review paper.

Point 2.6:

As this is a review article, please include a systematic discussion on data collection, methods included, articles reviewed etc.

Response 2.6:

We apologize if our workflow was not clearly shown before. In this version, we included a new Figure S1 to support our rationale and complement the description on lines 94-131. In this new version we included references to support our finding a novel proposal for further experiments.

Point 2.7:

Page 14: The Figure on this page doesn’t have a number. 

Response 2.7:

We apologize for this mistake. We have improved the structure of the document, including figures and their legends.

Reviewer 3 Report

The authors embark on an exploratory journey through the complex landscape of autophagy and its relevance in cancer progression and treatment resistance. This topic is indeed timely, given the increasing attention autophagy has garnered within the oncology research community in recent years.

The omics-based approach taken by the authors has allowed for a novel perspective on this challenging issue. The discovery of three genes (TBC1D12, KERA, and TUBA3D) not previously associated with autophagy pathways in cancer demonstrates the robustness of the methodologies used and opens up new avenues for research in this field.

Considering the quality of this work in demonstrating the complexities of tumor heterogeneity and the potential implications on the expression of autophagy-related genes, I recommend this manuscript to be accepted in its current form.

Author Response

Reviewer #3

Point 3.1:

The authors embark on an exploratory journey through the complex landscape of autophagy and its relevance in cancer progression and treatment resistance. This topic is indeed timely, given the increasing attention autophagy has garnered within the oncology research community in recent years.

The omics-based approach taken by the authors has allowed for a novel perspective on this challenging issue. The discovery of three genes (TBC1D12, KERA, and TUBA3D) not previously associated with autophagy pathways in cancer demonstrates the robustness of the methodologies used and opens up new avenues for research in this field.

Considering the quality of this work in demonstrating the complexities of tumor heterogeneity and the potential implications on the expression of autophagy-related genes, I recommend this manuscript to be accepted in its current form.

Response 3.1:

Thank you for this summary of our study and for emphasizing the relevance of analyzing autophagy genes in cancer.

Round 2

Reviewer 1 Report

No further comments after the authors addressed the previously mentioned topics.

Author Response

Thank you for your help and suggestions.

Reviewer 2 Report

Thanks for revisions. I still think the manuscript needs significant structural revisions. Please have a look at my following comments-

1) Is there any separate "method" section in the manuscript? I haven't found any. I would suggest keeping methodology under that heading. Though it is a review article, but authors have done plenty of data analyses. Thus, it becomes important to have the method section separately. Same for separate "Result" or "Result and discussion" sections. All results and discussions should go to these sections. It will be easier for readers. Please verify with the journal's format, too (https://www.mdpi.com/journal/genes/instructions#preparation). To me, this manuscript looks more like a full article, not a review article. Is there any reason why it was submitted as a review?

2) Point 2.2:  I suggested interpreting contribution proportion of genes TBC1D12, KERA, and TUBA3D for cluster prediction. Please explore Random Forest's gini importance method for this. As you claimed that these are highly influential genes. I believe adding this section with importance score analysis will be informative.

3) Line 116, remove duplicate "TCGA."

Author Response

As mentioned by the reviewer, there was no separate section on methods in our manuscript. We appreciate the reviewer's suggestion and in order to clarify this topic, we have added a described the methods in section 2. Likewise, we added the topic results and discussion, section 3. We kept results and discussion together to facilitate the understanding of the data by the reader and not to extend the text with repeated information.

Regarding topic 2, we thank you for the suggestion. We performed the analysis using random forest's gini to interpret the contribution of the 3 genes cited.